# Minimal Exploration
# in Structured Stochastic Bandits

**Richard Combes**
Centrale-Supelec / L2S
richard.combes@supelec.fr

**Stefan Magureanu**
KTH, EE School / ACL
magur@kth.se

**Alexandre Proutiere**
KTH, EE School / ACL
alepro@kth.se

## Abstract

This paper introduces and addresses a wide class of stochastic bandit problems where the function mapping the arm to the corresponding reward exhibits some known structural properties. Most existing structures (e.g. linear, Lipschitz, unimodal, combinatorial, dueling, ...) are covered by our framework. We derive an asymptotic instance-specific regret lower bound for these problems, and develop OSSB, an algorithm whose regret matches this fundamental limit. OSSB is not based on the classical principle of "optimism in the face of uncertainty" or on Thompson sampling, and rather aims at matching the minimal exploration rates of sub-optimal arms as characterized in the derivation of the regret lower bound. We illustrate the efficiency of OSSB using numerical experiments in the case of the linear bandit problem and show that OSSB outperforms existing algorithms, including Thompson sampling.

## 1 Introduction

Numerous extensions of the classical stochastic MAB problem [30] have been recently investigated. These extensions are motivated by applications arising in various fields including e.g. on-line services (search engines, display ads, recommendation systems, ...), and most often concern structural properties of the mapping of arms to their average rewards. This mapping can for instance be linear [14], convex [2], unimodal [36], Lipschitz [3], or may exhibit some combinatorial structure [10, 29, 35].

In their seminal paper, Lai and Robbins [30] develop a comprehensive theory for MAB problems with unrelated arms, i.e., without structure. They derive asymptotic (as the time horizon grows large) instance-specific regret lower bounds and propose algorithms achieving this minimal regret. These algorithms have then been considerably simplified, so that today, we have a few elementary index-based[1] and yet asymptotically optimal algorithms [18, 26]. Developing a similar comprehensive theory for MAB problems *with* structure is considerably more challenging. Due to the structure, the rewards observed for a given arm actually provide side-information about the average rewards of other arms[2]. This side-information should be exploited so as to accelerate as much as possible the process of learning the average rewards. Very recently, instance-specific regret lower bounds and asymptotically optimal algorithms could be derived only for a few MAB problems with finite set of arms and specific structures, namely linear [31], Lipschitz [32] and unimodal [12].

In this paper, we investigate a large class of structured MAB problems. This class extends the classical stochastic MAB problem [30] in two directions: (i) it allows for *any* arbitrary structure; (ii) it allows different kinds of feedback. More precisely, our generic MAB problem is as follows.

In each round, the decision maker selects an arm from a finite set $\mathcal{X}$. Each arm $x \in \mathcal{X}$ has an unknown parameter $\theta(x) \in \mathbb{R}$, and when this arm is chosen in round $t$, the decision maker observes a real-valued random variable $Y(x, t)$ with expectation $\theta(x)$ and distribution $\nu(\theta(x))$. The observations $(Y(x, t))_{x \in \mathcal{X}, t \geq 1}$ are independent across arms and rounds. If $x$ is chosen, she also receives an unobserved and deterministic[3] reward $\mu(x, \theta)$, where $\theta = (\theta(x))_{x \in \mathcal{X}}$. The parameter $\theta$ lies in a compact set $\Theta$ that encodes the structural properties of the problem. The set $\Theta$, the class of distributions $\nu$, and the mapping $(x, \theta) \mapsto \mu(x, \theta)$ encode the structure of the problem, are known to the decision maker, whereas $\theta$ is initially unknown. We denote by $x^\pi(t)$ the arm selected in round $t$ under algorithm $\pi$; this selection is based on previously selected arms and the corresponding observations. Hence the set $\Pi$ of all possible arm selection rules consists in algorithms $\pi$ such that for any $t \geq 1$, $x^\pi(t)$ is $\mathcal{F}_t^\pi$-measurable where $\mathcal{F}_t^\pi$ is the $\sigma$-algebra generated by $(x^\pi(1), Y(x^\pi(1), 1), \ldots, x^\pi(t-1), Y(x^\pi(t-1), t-1)$. The performance of an algorithm $\pi \in \Pi$ is defined through its regret up to round $T$:

$$R^\pi(T, \theta) = T \max_{x \in \mathcal{X}} \mu(x, \theta) - \sum_{t=1}^{T} \mathbb{E}(\mu(x(t), \theta)).$$

The above MAB problem is very generic, as any kind of structure can be considered. In particular, our problem includes classical, linear, unimodal, dueling, and Lipschitz bandit problems as particular examples, see Section 3 for details. Our contributions in this paper are as follows:

- We derive a tight instance-specific regret lower bound satisfied by any algorithm for our generic structured MAB problem.

- We develop OSSB (Optimal Sampling for Structured Bandits), a simple and yet asymptotically optimal algorithm, i.e., its regret matches our lower bound. OSSB optimally exploits the structure of the problem so as to minimize regret.

- We briefly exemplify the numerical performance of OSSB in the case of linear bandits. OSSB outperforms existing algorithms (including Thompson Sampling [2], GLM-UCB [16], and a recently proposed asymptotically optimal algorithm [31]).

As noticed in [31], for structured bandits (even for linear bandits), no algorithm based on the principle of optimism (a la UCB) or on that of Thompson sampling can achieve an asymptotically minimal regret. The design of OSSB does not follow these principles, and is rather inspired by the derivation of the regret lower bound. To obtain this bound, we characterize the minimal rates at which sub-optimal arms have to be explored. OSSB aims at sampling sub-optimal arms so as to match these rates. The latter depends on the unknown parameter $\theta$, and so OSSB needs to accurately estimate $\theta$. OSSB hence alternates between three phases: exploitation (playing arms with high empirical rewards), exploration (playing sub-optimal arms at well chosen rates), and estimation (getting to know $\theta$ to tune these exploration rates).

The main technical contribution of this paper is a finite-time regret analysis of OSSB for any generic structure. In spite of the simplicity the algorithm, its analysis is involved. Not surprisingly, it uses concentration-of-measure arguments, but it also requires to establish that the minimal exploration rates (derived in the regret lower bound) are essentially smooth with respect to the parameter $\theta$. This complication arises due to the (additional) estimation phase of OSSB: the minimal exploration rates should converge as our estimate of $\theta$ gets more and more accurate.

The remainder of the paper is organized as follows. In the next section, we survey recent results on structured stochastic bandits. In Section 3, we illustrate the versatility of our MAB problem by casting most existing structured bandit problems into our framework. Section 4 is devoted to the derivation of the regret lower bound. In Sections 5 and 6, we present OSSB and provide an upper bound of its regret. Finally Section 7 explores the numerical performance of OSSB in the case of linear structures.

## 2 Related work

Structured bandits have generated many recent contributions since they find natural applications in the design of computer systems, for instance: recommender systems and information retrieval [28, 11], routing in networks and network optimization [22, 5, 17], and influence maximization in social networks [8]. A large number of existing structures have been investigated, including: linear [14, 34, 1, 31, 27] (linear bandits are treated here as a partial monitoring game), combinatorial [9, 10, 29, 35, 13], Lipschitz [32], unimodal [36, 12]. The results in this paper cover all models considered in the above body of work and are the first that can be applied to problems with any structure in the set of allowed parameters.

Here, we focus on generic stochastic bandits with a finite but potentially large number of arms. Both continuous as well as adversarial versions of the problem have been investigated, see survey [6].

The performance of Thompson sampling for generic bandit problems has appeared in the literature [15, 20], however, the recent results in [31] prove that Thompson sampling is not optimal for all structured bandits. Generic structured bandits were treated in [7, 21]. The authors show that the regret of any algorithm must scale as $C(\theta)\ln T$ when $T \to \infty$ where $C(\theta)$ is the optimal value of a semi-infinite linear program, and propose asymptotically optimal algorithms. However the proposed algorithms are involved and have poor numerical performance, furthermore their performance guarantees are asymptotic, and no finite time analysis is available.

To our knowledge, our algorithm is the first which covers completely generic MAB problems, is asymptotically optimal and is amenable to a finite-time regret analysis. Our algorithm is in the same spirit as the DMED algorithm, presented in [24], as well as the algorithm in [31], but is generic enough to be optimal in any structured bandit setting. Similar to DMED, our algorithm relies on repeatedly solving an optimization problem and then exploring according to its solution, thus moving away from the UCB family of algorithms.

## 3 Examples

The class of MAB problems described in the introduction covers most known bandit problems as illustrated in the six following examples.

**Classical Bandits.** The classical MAB problem [33] with Bernoulli rewards is obtained by making the following choices: $\theta(x) \in [0,1]$; $\Theta = [0,1]^{|\mathcal{X}|}$; for any $a \in [0,1]$, $\nu(a)$ is the Bernoulli distribution with mean $a$; for all $x \in \mathcal{X}$, $\mu(x,\theta) = \theta(x)$.

**Linear Bandits.** To get finite linear bandit problems [14],[31], in our framework we choose $\mathcal{X}$ as a finite subset of $\mathbb{R}^d$; we pick an unknown vector $\phi \in \mathbb{R}^d$ and define $\theta(x) = \langle \phi, x \rangle$ for all $x \in \mathcal{X}$; the set of possible parameters is $\Theta = \{\theta = (\langle \phi, x \rangle)_{x \in \mathcal{X}}, \phi \in \mathbb{R}^d\}$; for any $a \in \mathbb{R}^d$, $\nu(a)$ is a Gaussian distribution with unit variance and centered at $a$; for all $x \in \mathcal{X}$, $\mu(x,\theta) = \theta(x)$. Observe that our framework also includes generalized linear bandit problems as those considered in [16]: we just need to define $\mu(x,\theta) = g(\theta(x))$ for some function $g$.

**Dueling Bandits.** To model dueling bandits [27] using our framework, the set of arms is $\mathcal{X} = \{(i,j) \in \{1,\ldots,d\}^2\}$; for any $x = (i,j) \in \mathcal{X}$, $\theta(x) \in [0,1]$ denotes the probability that $i$ is *better* than $j$ with the conventions that $\theta(i,j) = 1 - \theta(j,i)$ and that $\theta(i,i) = 1/2$; $\Theta = \{\theta : \exists i^\star : \theta(i^\star, j) > 1/2, \forall j \neq i^\star\}$ is the set of parameters such there exists a Condorcet winner; for any $a \in [0,1]$, $\nu(a)$ is the Bernoulli distribution with mean $a$; finally, we define the rewards as $\mu((i,j),\theta) = \frac{1}{2}(\theta(i^\star,i) + \theta(i^\star,j) - 1)$. Note that the best arm is $(i^\star, i^\star)$ and has zero reward.

**Lipschitz Bandits.** For finite Lipschitz bandits [32], the set of arms $\mathcal{X}$ is a finite subset of a metric space endowed with a distance $\ell$. For any $x \in \mathcal{X}$, $\theta(x)$ is a scalar, and the mapping $x \mapsto \theta(x)$ is Lipschitz continuous with respect to $\ell$, and the set of parameters is:

$$\Theta = \{\theta : |\theta(x) - \theta(y)| \leq \ell(x,y) \quad \forall x, y \in \mathcal{X}\}.$$

As in classical bandits $\mu(x,\theta(x)) = \theta(x)$. The structure is encoded by the distance $\ell$, and is an example of local structure so that arms close to each other have similar rewards.

**Unimodal Bandits.** Unimodal bandits [23],[12] are obtained as follows. $\mathcal{X} = \{1, ..., |\mathcal{X}|\}$, $\theta(x)$ is a scalar, and $\mu(x,\theta(x)) = \theta(x)$. The added assumption is that $x \mapsto \theta(x)$ is unimodal. Namely, there

exists $x^\star \in \mathcal{X}$ such that this mapping is stricly incrasing on $\{1, ..., x^\star\}$ and strictly decreasing on $\{x^\star, ..., |\mathcal{X}|\}$.

**Combinatorial bandits.** The combinatorial bandit problems with *bandit feedback* (see [9]) are just particular instances of linear bandits where the set of arms $\mathcal{X}$ is a subset of $\{0, 1\}^d$. Now to model combinatorial problems with *semi-bandit* feedback, we need a slight extension of the framework described in introduction. More precisely, the set of arms is still a subset of $\{0, 1\}^d$. The observation $Y(x, t)$ is a $d$-dimensional r.v. with independent components, with mean $\theta(x)$ and distribution $\nu(\theta(x))$ (a product distribution). There is an unknown vector $\phi \in \mathbb{R}^d$ such that $\theta(x) = (\phi(1)x(1), \ldots, \phi(d)x(d))$, and $\mu(x, \theta) = \sum_{i=1}^{d} \phi(i)x(i)$ (linear reward). With semi-bandit feedback, the decision maker gets detailed information about the various components of the selected arm.

## 4 Regret Lower Bound

To derive regret lower bounds, a strategy consists in restricting the attention to so-called *uniformly good* algorithms [30]: $\pi \in \Pi$ is uniformly good if $R^\pi(T, \theta) = o(T^a)$ when $T \to \infty$ for all $a > 0$ and all $\theta \in \Theta$. A simple change-of-measure argument is then enough to prove that for MAB problems without structure, under any uniformly good algorithm, the number of times that a sub-optimal arm $x$ should be played is greater than $\ln T / d(\theta(x), \theta(x^\star))$ as the time horizon $T$ grows large, and where $x^\star$ denotes the optimal arm and $d(\theta(x), \theta(x^\star))$ is the Kullback-Leibler divergence between the distributions $\nu(\theta(x))$ and $\nu(\theta(x^\star))$. Refer to [25] for a direct and elegant proof.

For our structured MAB problems, we follow the same strategy, and derive constraints on the number of times a sub-optimal arm $x$ is played under any uniformly good algorithm. We show that this number is greater than $c(x, \theta) \ln T$ asymptotically where the $c(x, \theta)$'s are the solutions of a semi-infinite linear program [19] whose constraints directly depend on the structure of the problem.

Before stating our lower bound, we introduce the following notations. For $\theta \in \Theta$, let $x^\star(\theta)$ be the optimal arm (we assume that it is unique), and define $\mu^\star(\theta) = \mu(x^\star(\theta), \theta)$. For any $x \in \mathcal{X}$, we denote by $D(\theta, \lambda, x)$ the Kullback-Leibler divergence between distributions $\nu(\theta(x))$ and $\nu(\lambda(x))$.

**Assumption 1** *The optimal arm $x^\star(\theta)$ is unique.*

**Theorem 1** *Let $\pi \in \Pi$ be a uniformly good algorithm. For any $\theta \in \Theta$, we have:*

$$\liminf_{T \to \infty} \frac{R^\pi(T, \theta)}{\ln T} \geq C(\theta),\tag{1}$$

*where $C(\theta)$ is the value of the optimization problem:*

$$\underset{\eta(x) \geq 0\, ,\, x \in \mathcal{X}}{minimize} \sum_{x \in \mathcal{X}} \eta(x)(\mu^\star(\theta) - \mu(x, \theta))\tag{2}$$

$$subject\ to \sum_{x \in \mathcal{X}} \eta(x)D(\theta, \lambda, x) \geq 1\, ,\, \forall \lambda \in \Lambda(\theta),\tag{3}$$

*where*

$$\Lambda(\theta) = \{\lambda \in \Theta : D(\theta, \lambda, x^\star(\theta)) = 0, x^\star(\theta) \neq x^\star(\lambda)\}.\tag{4}$$

Let $(c(x, \theta))_{x \in \mathcal{X}}$ denote the solutions of the semi-infinite linear program (2)-(3). In this program, $\eta(x) \ln T$ indicates the number of times arm $x$ is played. The regret lower bound may be understood as follows. The set $\Lambda(\theta)$ is the set of "confusing" parameters: if $\lambda \in \Lambda(\theta)$ then $D(\theta, \lambda, x^\star(\theta)) = 0$ so $\lambda$ and $\theta$ cannot be differentiated by only sampling the optimal arm $x^\star(\theta)$. Hence distinguishing $\theta$ from $\lambda$ requires to sample suboptimal arms $x \neq x^\star(\theta)$. Further, since any uniformly good algorithm must identify the best arm with high probability to ensure low regret and $x^\star(\theta) \neq x^\star(\lambda)$, any algorithm must distinguish these two parameters. The constraint (3) states that for any $\lambda$, a uniformly good algorithm should perform a hypothesis test between $\theta$ and $\lambda$, and $\sum_{x \in \mathcal{X}} \eta(x)D(\theta, \lambda, x) \geq 1$ is required to ensure there is enough statistical information to perform this test. In summary, for a sub-optimal arm $x$, $c(x, \theta) \ln T$ represents the asymptotically minimal number of times $x$ should be sampled. It is noted that this lower bound is instance-specific (it depends on $\theta$), and is attainable as we propose an algorithm which attains it. The proof of Theorem 1 is presented in appendix, and leverages techniques used in the context of controlled Markov chains [21].

Next, we show that with usual structures as those considered in Section 3, the semi-infinite linear program (2)-(3) reduces to simpler optimization problems (e.g. an LP) and can sometimes even be solved explicitly. Simplifying (2)-(3) is important for us, since our proposed asymptotically optimal algorithm requires to solve this program. In the following examples, please refer to Section 3 for the definitions and notations. As mentioned already, the solutions of (2)-(3) for classical MAB is $c(x, \theta) = 1/d(\theta(x), \theta(x^\star))$.

**Linear bandits.** For this class of problems, [31] recently proved that (2)-(3) was equivalent to the following optimization problem:

$$\underset{\eta(x) \geq 0\,,\, x \in \mathcal{X}}{\text{minimize}} \sum_{x \in \mathcal{X}} \eta(x)(\theta(x^\star) - \theta(x))$$

$$\text{subject to } x^\top \mathbf{inv}\left(\sum_{z \in \mathcal{X}} \eta(z)zz^\top\right) x \leq \frac{(\theta(x^\star) - \theta(x))^2}{2},$$

$$\forall x \neq x^\star.$$

Refer to [31] for the proof of this result, and for insightful discussions.

**Lipschitz bandits.** It can be shown that for Bernoulli rewards (the reward of arm $x$ is $\theta(x)$) (2)-(3) reduces to the following LP [32]:

$$\underset{\eta(x) \geq 0\,,\, x \in \mathcal{X}}{\text{minimize}} \sum_{x \in \mathcal{X}} \eta(x)(\theta(x^\star) - \theta(x))$$

$$\text{subject to } \sum_{z \in \mathcal{X}} \eta(z)d(\theta(z), \max\{\theta(z), \theta(x^\star) - \ell(x, z)\}) \geq 1\,,$$

$$\forall x \neq x^\star.$$

While the solution is not explicit, the problem reduces to a LP with $|\mathcal{X}|$ variables and $2|\mathcal{X}|$ constraints.

**Dueling bandits.** The solution of (2)-(3) is as follows [27]. Assume to simplify that for any $i \neq i^\star$, there exists a unique $j$ minimizing $\frac{\mu((i,j),\theta)}{d(\theta(i,j),1/2)}$ and such that $\theta(i, j) < 1/2$. Let $j(i)$ denote this index. Then for any $x = (i, j)$, we have

$$c(x, \theta) = \frac{\mathbf{1}\{j = j(i)\}}{d(\theta(i, j), 1/2)}.$$

**Unimodal bandits.** For such problems, it is shown in [12] that the solution of (2)-(3) is given by: for all $x \in \mathcal{X}$,

$$c(x, \theta) = \frac{\mathbf{1}\{|x - x^\star| = 1\}}{d(\theta(x), \theta(x^\star))}.$$

Hence, in unimodal bandits, under an asymptotically optimal algorithm, the sub-optimal arms contributing to the regret (i.e., those that need to be sampled $\Omega(\ln T)$) are neighbours of the optimal arm.

## 5 The OSSB Algorithm

In this section we propose OSSB (Optimal Sampling for Structured Bandits), an algorithm that is asymptotically optimal, i.e., its regret matches the lower bound of Theorem 1. OSSB pseudo-code is presented in Algorithm 1, and takes as an input two parameters $\varepsilon, \gamma > 0$ that control the amount of exploration performed by the algorithm.

The design of OSSB is guided by the necessity to explore suboptimal arms as much as prescribed by the solution of the optimization problem (2)-(3), i.e., the sub-optimal arm $x$ should be explored $c(x, \theta)\ln T$ times. If $\theta$ was known, then sampling arm $x$ $c(x, \theta)\ln T$ times for all $x$, and then selecting the arm with the largest estimated reward should yield minimal regret.

Since $\theta$ is unknown, we have to estimate it. Define the empirical averages:

$$m(x, t) = \frac{\sum_{s=1}^{t} Y(x, s)\mathbf{1}\{x(s) = x\}}{\max(1, N(x, t))}$$

---

**Algorithm 1** OSSB($\varepsilon,\gamma$)

---

$s(0) \leftarrow 0, N(x,1), m(x,1) \leftarrow 0 , \forall x \in \mathcal{X}$          {Initialization}

**for** $t = 1, ..., T$ **do**

     Compute the optimization problem (2)-(3) solution $(c(x, m(t)))_{x \in \mathcal{X}}$ where $m(t) = (m(x,t))_{x \in \mathcal{X}}$

     **if** $N(x,t) \geq c(x, m(t))(1 + \gamma)\ln t, \forall x$ **then**

         $s(t) \leftarrow s(t-1)$

         $x(t) \leftarrow x^\star(m(t))$          {Exploitation}

     **else**

         $s(t) \leftarrow s(t-1) + 1$

         $\overline{X}(t) \leftarrow \arg\min_{x \in \mathcal{X}} \frac{N(x,t)}{c(x,m(t))}$

         $\underline{X}(t) \leftarrow \arg\min_{x \in \mathcal{X}} N(x,t)$

         **if** $N(\underline{X}(t), t) \leq \varepsilon s(t)$ **then**

             $x(t) \leftarrow \underline{X}(t)$          {Estimation}

         **else**

             $x(t) \leftarrow \overline{X}(t)$          {Exploration}

         **end if**

     **end if**

     {Update statistics}

     Select arm $x(t)$ and observe $Y(x(t), t)$

     $m(x, t+1) \leftarrow m(x,t), \forall x \neq x(t)$ ,

     $N(x, t+1) \leftarrow N(x,t), \forall x \neq x(t)$

     $m(x(t), t+1) \leftarrow \frac{Y(x(t),t) + m(x(t),t)N(x(t),t)}{N(x(t),t)+1}$

     $N(x(t), t+1) \leftarrow N(x(t),t) + 1$

**end for**

---

where $x(s)$ is the arm selected in round $s$, and $N(x,t) = \sum_{s=1}^{t} \mathbf{1}\{x(s) = x\}$ is the number of times $x$ has been selected up to round $t$. The key idea of OSSB is to use $m(t) = (m(x,t))_{x \in \mathcal{X}}$ as an estimator for $\theta$, and explore arms to match the *estimated* solution of the optimization problem (2)-(3), so that $N(x,t) \approx c(x, m(t))\ln t$ for all $x$. This should work if we can ensure *certainty equivalence*, i.e. $m(t) \to \theta(t)$ when $t \to \infty$ at a sufficiently fast rate.

The OSSB algorithm has three components. More precisely, under OSSB, we alternate between three phases: exploitation, estimation and exploration. In round $t$, one first attempts to identify the optimal arm. We calculate $x^\star(m(x,t))$ the arm with the largest empirical reward. If $N(x,t) \geq c(x, m(t))(1 + \gamma)\ln t$ for all $x$, we enter the *exploitation phase*: we have enough information to infer that $x^\star(m(x,t)) = x^\star(\theta)$ w.h.p. and we select $x(t) = x^\star(m(x,t))$. Otherwise, we need to gather more information to identify the optimal arm. We have two goals: (i) make sure that all components of $\theta$ are accurately estimated and (ii) make sure that $N(x,t) \approx c(x, m(t))\ln t$ for all $x$. We maintain a counter $s(t)$ of the number of times we have not entered the expoitation phase. We choose between two possible arms, namely the least played arm $\underline{X}(t)$ and the arm $\overline{X}(t)$ which is the farthest from satisfying $N(x,t) \geq c(x, m(t))\ln t$. We then consider the number of times $\underline{X}(t)$ has been selected. If $N(\underline{X}(t), t)$ is much smaller than $s(t)$, there is a possibility that $\underline{X}(t)$ has not been selected enough times so that $\theta(\underline{X}(t))$ is not accurately estimated so we enter the *estimation* phase, where we select $\underline{X}(t)$ to ensure that certainty equivalence holds. Otherwise we enter the *exploration* phase where we select $\overline{X}(t)$ to explore as dictated by the solution of (2)-(3), since $c(x, m(t))$ should be close to $c(x, \theta)$.

Theorem 2 states that OSSB is asymptotically optimal. The complete proof is presented in Appendix, with a sketch of the proof provided in the next section. We prove Theorem 2 for Bernoulli or Subgaussian observations, but the analysis is easily extended to rewards in a 1-parameter exponential family of distributions. While we state an asymptotic result here, we actually perform a finite time analysis of OSSB, and a finite time regret upper bound for OSSB is displayed at the end of next section.

**Assumption 2** *(Bernoulli observations)* $\theta(x) \in [0, 1]$ *and* $\nu(\theta(x)) = Ber(\theta(x))$ *for all* $x \in \mathcal{X}$.

**Assumption 3** *(Gaussian observations)* $\theta(x) \in \mathbb{R}$ *and* $\nu(\theta(x)) = \mathcal{N}(\theta(x), 1)$ *for all* $x \in \mathcal{X}$.

**Assumption 4** *For all $x$, the mapping $(\theta, \lambda) \mapsto D(x, \theta, \lambda)$ is continuous at all points where it is not infinite.*

**Assumption 5** *For all $x$, the mapping $\theta \to \mu(x, \theta)$ is continuous.*

**Assumption 6** *The solution to problem (2)-(3) is unique.*

**Theorem 2** *If Assumptions 1, 4, 5 and 6 hold and either Assumption 2 or 3 holds, then under the algorithm $\pi =$ OSSB$(\varepsilon, \gamma)$ with $\varepsilon < \frac{1}{|\mathcal{X}|}$ we have:*

$$\limsup_{T \to \infty} \frac{R^\pi(T)}{\ln T} \leq C(\theta) F(\varepsilon, \gamma, \theta),$$

*with $F$ a function such that for all $\theta$, we have $F(\varepsilon, \gamma, \theta) \to 1$ as $\varepsilon \to 0$ and $\gamma \to 0$.*

We conclude this section by a remark on the computational complexity of the OSSB algorithm. OSSB requires to solve the optimization problem (2)-(3) in each round. The complexity of solving this problem strongly depends on the problem structure. For general structures, the complexity of this problem is difficult to assess. However for problems exemplified in Section 3, this problem is usually easy to solve. Note that the algorithm proposed in [31] for linear bandits requires to solve (2)-(3) only once, and is hence simpler to implement; its performance however is much worse in practice than that of OSSB as illustrated in Section 7.

## 6   Finite Time Analysis of OSSB

The proof of Theorem 2 is presented in Appendix in detail, and is articulated in four steps. (i) We first notice that the probability of selecting a suboptimal arm during the exploitation phase at some round $t$ is upper bounded by $\mathbb{P}(\sum_{x \in \mathcal{X}} N(x, t) D(m(t), \theta, x) \geq (1 + \gamma) \ln t)$. Using a concentration inequality on KL-divergences (Lemma 1 in Appendix), we show that this probability is small and the regret caused by the exploitation phase is upper bounded by $G(\gamma, |\mathcal{X}|)$ where $G$ is finite and depends solely on $\gamma$ and $|\mathcal{X}|$. (ii) The second step, which is the most involved, is to show Lemma 1 stating the solutions of (2)-(3) are *continuous*. The main difficulty is that the set $\Lambda(\theta)$ is not finite, so that the optimization problem (2)-(3) is not a linear program. The proof strategy is similar to that used to prove Berge's maximal theorem, the additional difficulty being that the feasible set is not compact, so that Berge's theorem cannot be applied directly. Using Assumptions 1 and 5, both the value $\theta \mapsto C(\theta)$ and the solution $\theta \mapsto c(\theta)$ are continuous.

**Lemma 1** *The optimal value of (2)-(3), $\theta \mapsto C(\theta)$ is continuous. If (2)-(3) admits a unique solution $c(\theta) = (c(x, \theta))_{x \in \mathcal{X}}$ at $\theta$, then $\theta \mapsto c(\theta)$ is continuous at $\theta$.*

Lemma 1 is in fact interesting in its own right, since optimization problems such as (2)-(3) occur in *all* bandit problems. (iii) The third step is to upper bound the number of times the solution to (2)-(3) is not well estimated, so that $C(m(t)) \geq (1 + \kappa)C(\theta)$ for some $\kappa > 0$. From the previous step this implies that $||m(t) - \theta||_\infty \geq \delta(\kappa)$ for some well-chosen $\delta(\kappa) > 0$. Using a deviation result (Lemma 2 in Appendix), we show that the expected regret caused by such events is finite and upper bounded by $\frac{2|\mathcal{X}|}{\varepsilon \delta^2(\kappa)}$. (vi) Finally a counting argument ensures that the regret incurred when $C(\theta) \leq C(m(t)) \leq (1 + \kappa)C(\theta)$ i.e. the solution (2)-(3) is well estimated is upper bounded by $(C(\theta)(1 + \kappa) + 2\varepsilon\psi(\theta))\ln T$, where $\psi(\theta) = |\mathcal{X}|||c(\theta)||_\infty \sum_{x \in \mathcal{X}} (\mu^\star(\theta) - \mu(x, \theta))$.

Putting everything together we obtain the finite-time regret upper bound:

$$R^\pi(T) \leq \mu^\star(\theta) \left( G(\gamma, |\mathcal{X}|) + \frac{2|\mathcal{X}|}{\varepsilon \delta^2(\kappa)} \right) + (C(\theta)(1 + \kappa) + 2\varepsilon\psi(\theta))(1 + \gamma)\ln T.$$

This implies that:

$$\limsup_{T \to \infty} \frac{R^\pi(T)}{\ln T} \leq (C(\theta)(1 + \kappa) + 2\varepsilon\psi(\theta))(1 + \gamma).$$

The above holds for all $\kappa > 0$, which yields the result.

# 7 Numerical Experiments

To assess the efficiency of OSSB, we compare its performance for reasonable time horizons to the state of the art algorithms for linear bandit problems. We considered a linear bandit with Gaussian rewards of unit variance, $81$ arms of unit length, $d = 3$ and $10$ parameters $\theta$ in $[0.2, 0.4]^3$, generated uniformly at random. In our implementation of OSSB, we use $\gamma = \varepsilon = 0$ since $\gamma$ is typically chosen $0$ in the literature (see [18]) and the performance of the algorithm does not appear sensitive to the choice of $\varepsilon$. As baselines we select the extension of Thompson Sampling presented in [4](using $v_t = R\sqrt{0.5d\ln(t/\delta)}$, we chose $\delta = 0.1$, $R = 1$), GLM-UCB (using $\rho(t) = \sqrt{0.5\ln(t)}$), an extension of UCB [16] and the algorithm presented in [31].

Figure 1 presents the regret of the various algorithms averaged over the 10 parameters. OSSB clearly exhibits the best performance in terms of average regret.

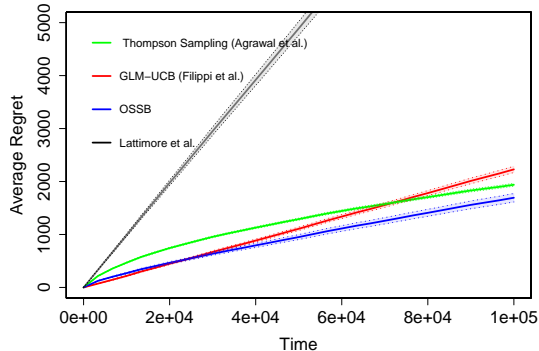

Figure 1: Regret of various algorithms in the linear bandit setting with $81$ arms and $d = 3$. Regret is averaged over 10 randomly generated parameters and 100 trials. Colored regions represent the $95\%$ confidence intervals.

# 8 Conclusion

In this paper, we develop a unified solution to a wide class of stochastic structured bandit problems. For the first time, we derive, for these problems, an asymptotic regret lower bound and devise OSSB, a simple and yet asymptotically optimal algorithm. The implementation of OSSB requires that we solve the optimization problem defining the minimal exploration rates of the sub-optimal arms. In the most general case, this problem is a semi-infinite linear program, which can be hard to solve in reasonable time. Studying the complexity of this semi-infinite LP depending on the structural properties of the reward function is an interesting research direction. Indeed any asymptotically optimal algorithm needs to learn the minimal exploration rates of sub-optimal arms, and hence needs to solve this semi-infinite LP. Characterizing the complexity of the latter would thus yield important insights into the trade-off between the complexity of the sequential arm selection algorithms and their regret.

### Acknowledgments

A. Proutiere's research is supported by the ERC FSA (308267) grant. This work is supported by the French Agence Nationale de la Recherche (ANR), under grant ANR-16-CE40-0002 (project BADASS).

## Footnotes

[1]An algorithm is *index-based* if the arm selection in each round is solely made comparing the indexes of each arm, and where the index of an arm only depends on the rewards observed for this arm.

[2]Index-based algorithms cannot be optimal in MAB problems with structure.

[3]Usually in MAB problems, the reward is a random variable given as feedback to the decision maker. In our model, the reward is deterministic (as if it was averaged), but not observed as the only observation is $Y(x, t)$ if $x$ is chosen in round $t$. We will illustrate in Section 3 why usual MAB formulations are specific instances of our model.

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
