[Supplementary Material]

# Minimal Exploration in Structured Stochastic Bandits Supplementary material

**Richard Combes**
Centrale-Supelec / L2S
richard.combes@supelec.fr

**Stefan Magureanu**
KTH, EE School / ACL
magur@kth.se

**Alexandre Proutiere**
KTH, EE School / ACL
alepro@kth.se

## 1   Proof of Theorem 1.

We first recall the original framework presented in [2], whose results we apply to establish the asymptotic lower bound in Theorem 1. Consider a controlled Markov chain $(Y_t)_{t \geq 0}$ on a measurable state space $\mathcal{Y}$ with a control set $U$. For a given control $u \in U$ the transition probabilities are parametrized by $\theta \in \Theta$, where $\Theta$ is a compact metric space. We denote the probability to move from state $y$ to state $y'$, given the control $u$ and the parameter $\theta$ as $p(y, y'|u, \theta)$. The parameter $\theta$ is not known. The decision maker is provided with a finite set of stationary control laws $G = \{g_1, \ldots, g_K\}$ where each control law $g_j$ is a mapping from $\mathcal{Y}$ to $U$: when control law $g_j$ is applied in state $y$, the applied control is $u = g_j(y)$. It is assumed that if the decision maker always selects the same control law $g$, the Markov chain is ergodic with stationary distribution $\pi_\theta^g$. The expected reward obtained when applying control $u$ in state $y$ is denoted by $r(y, u)$, so that the expected reward achieved under control law $g$ is: $\mu(\theta, g) = \int r(y, g(y)) \pi_\theta^g(y) dy$. The decision maker knows the mapping $r$ but not $\theta$, and she selects control laws $g(1), g(2), \ldots$, to minimize the cumulated regret:

$$R(T, \theta) = \sum_{t=1}^{T} \max_{g \in G} \mu(\theta, g) - \mu(\theta, g(t)).$$

The chosen control law at time $t$ solely depends on the observed states $Y_1, \ldots, Y_t$ and the past chosen control laws $g(1), \ldots, g(t-1)$. It should be noted that this framework, and the corresponding results, can be straightforwardly extended to a case where the mapping $(\theta, g) \mapsto \mu(\theta, g)$ is arbitrary, as long as this mapping is known to the decision maker. Of course $\theta$ remains unknown.

We now apply the above framework to our structured bandit problem, and we consider $\theta \in \Theta$, the compact metric space encoding the structural properties of the average reward function (as introduced in Section 1). The Markov chain has values in $\mathcal{Y} = \mathbb{R}$. The set of control laws is $G = U = \mathcal{X}$, the set of available arms. These laws are constant, in the sense that the control applied by control law $x$ does not depend on the state of the Markov chain, and corresponds to selecting arm $x$. The state of the Markov chain at time $t$ is given by the observation $Y_t = Y(x(t-1), t-1)$. The transition probabilities are chosen such that when control law $x$ is chosen, $Y_{t+1}$ is distributed as $\nu(\theta(x))$, independently of $Y_t$.

Therefore the Kullback-Leibler information number in the framework of [2] is simply the Kullback-Leibler divergence between the distributions $\nu(\theta(x))$ and $\nu(\lambda(x))$, that is $I^x(\theta, \lambda) = D(\theta, \lambda, x)$.

Now consider $\theta \in \Theta$ and define the set $\Lambda(\theta)$ consisting of all *confusing* parameters $\lambda \in \Theta$ such that $x^\star(\theta)$ is not optimal under parameter $\lambda$, but which are statistically *indistinguishable* from $\theta$ when playing only $x^\star(\theta)$:

$$\Lambda(\theta) = \{\lambda \in \Theta : D(\theta, \lambda, x^\star(\theta)) = 0, x^\star(\theta) \neq x^\star(\lambda)\}$$

By applying Theorem 1 in [2], we know that for all *uniformly good* decision policies $\pi$ we have $\liminf_{T \to \infty} R^\pi(T, \theta) / \ln T \leq C(\theta)$, where $C(\theta)$ is the minimal value of the following optimization

problem:

$$\underset{\eta(x)\geq 0\,,\,x\in\mathcal{X}}{\text{minimize}}\sum_{x\in\mathcal{X}}\eta(x)(\mu^{\star}(\theta)-\mu(x,\theta))$$

$$\text{subject to}\sum_{x\in\mathcal{X}}\eta(x)D(\theta,\lambda,x)\geq 1\,,\,\forall\lambda\in\Lambda(\theta).$$

which concludes the proof. $\square$

## 2 Finite Time Analysis of OSSB

For completeness, we restate the assumptions on which the theorem relies:

**Assumption 1** *The optimal arm $x^{\star}(\theta)$ is unique.*

**Assumption 2** *(Bernoulli observations) $\theta(x)\in[0,1]$ and $\nu(\theta(x))=Ber(\theta(x))$ for all $x\in\mathcal{X}$.*

**Assumption 3** *(Gaussian observations) $\theta(x)\in\mathbb{R}$ and $\nu(\theta(x))=\mathcal{N}(\theta(x),1)$ for all $x\in\mathcal{X}$.*

**Assumption 4** *For all $x$, the mapping $(\theta,\lambda)\mapsto D(x,\theta,\lambda)$ is continuous at all points where it is not infinite.*

**Assumption 5** *For all $x$, the mapping $\theta\to\mu(x,\theta)$ is continuous.*

**Assumption 6** *The solution to problem (2)-(3) in Theorem 1 is unique.*

We now prove Theorem 2 and give a finite time analysis of OSSB.

### 2.1 Concentration results

We first state two technical results which are instrumental to our analysis.

**Lemma 1** *[3] Consider either Assumption 2 or 3. Then there exists a function G such that for all $t\geq 1$:*

$$\sum_{t\geq 1}\mathbb{P}[\sum_{x\in\mathcal{X}}N(x,t)D(m(t),\theta,x)\geq(1+\gamma)\ln t]\leq G(\gamma,|\mathcal{X}|).$$

**Lemma 2** *[1] Let $x\in\mathcal{X}$ and $\epsilon>0$. Define $\mathcal{F}_t$ the $\sigma$-algebra generated by $(Y(x(s),s))_{1\leq s\leq t}$. Let $\mathcal{S}\subset\mathbb{N}$ be a (random) set of rounds. Assume that there exists a sequence of (random) sets $(\overline{\mathcal{S}}(s))_{s\geq 1}$ such that (i) $\mathcal{S}\subset\cup_{s\geq 1}\mathcal{S}(s)$, (ii) for all $s\geq 1$ and all $t\in\mathcal{S}(s)$, $N(x,t)\geq\epsilon s$, (iii) $|\mathcal{S}(s)|\leq 1$, and (iv) the event $t\in\mathcal{S}(s)$ is $\mathcal{F}_t$-measurable. Then for all $\delta>0$:*

$$\sum_{t\geq 1}\mathbb{P}(t\in\mathcal{S},|m(x,t)-\theta(x)|>\delta)\leq\frac{1}{\epsilon\delta^2}.$$

Lemma 1 was first proven in [3] to analyze Lipshitz bandits, but it is useful for generic bandit problems as well, and allows to bound the expected number of times that the optimal arm is not correctly identified if arms have been sampled enough to meet the constraints of the optimization problem (2)-(3). Lemma 2 was proven in [1] in the context of unimodal bandits. However it is versatile and allows to upper bound the expected cardinality of any set of rounds where $x$ is selected and $\theta(x)$ is not accurately estimated. As shown by this lemma, such sets of rounds only cause *finite* regret.

### 2.2 Continuity of the optimization problem (2)-(3) in Theorem 1

Lemma 3 states that the optimization problem (2)-(3) in Theorem 1 (henceforth referred to as only (2)-(3)) is continuous with respect to $\theta$. This fact is the cornerstone of our analysis. Since all bandit problems feature optimization problems such as the one we consider here, Lemma 3 seems interesting in its own right. The main difficulty to prove Lemma 3 comes from the fact that the set

$\Lambda(\theta)$ is not finite, so that the optimization problem (2)-(3) is not a linear program. The proof strategy is similar to that used to prove Berge's maximal theorem, the added difficulty being that the feasible set is not compact, so that Berge's theorem cannot be applied directly.

**Lemma 3** *The optimal value of (2)-(3), $\theta \mapsto C(\theta)$ is continuous. If (2)-(3) admits a unique solution $c(\theta) = (c(x, \theta))_{x \in \mathcal{X}}$ at $\theta$, then $\theta \mapsto c(\theta)$ is continuous at $\theta$.*

*Proof.* Define $\Delta(\theta, x) = \mu^\star(\theta) - \mu(x, \theta)$. To ease notation, we use a vector notation so that $c(\theta), D(\theta, \lambda)$ and $\Delta(\theta)$ denote vectors in $\mathbb{R}^{|\mathcal{X}|}$ whose respective components are $c(\theta, x)$, $D(\theta, \lambda, x)$ and $\Delta(\theta, x)$. For any $v \in \mathbb{R}^{|\mathcal{X}|}$ we define $||v||_\infty = \max_{x \in \mathcal{X}} |v(x)|$.

Now define the set:

$$\Lambda'(\theta) = \{\lambda \in \Theta : \max_{x \neq x^\star(\theta)} \mu(x, \lambda) > \mu(x^\star(\theta), \lambda)\}.$$

and the feasible sets:

$$\mathcal{F}(\theta) = \{c \in (\mathbb{R}^+)^{|\mathcal{X}|} : \inf_{\lambda \in \Lambda(\theta)} \langle c, D(\theta, \lambda)\rangle \geq 1\},$$

$$\mathcal{F}'(\theta) = \{c \in (\mathbb{R}^+)^{|\mathcal{X}|} : \inf_{\lambda \in \Lambda'(\theta)} \langle c, D(\theta, \lambda)\rangle \geq 1\}$$

So $C(\theta)$ is the minimum of $c \mapsto \langle c, \Delta(\theta)\rangle$ over $\mathcal{F}(\theta)$. Define $C'(\theta)$ the minimum of $c \mapsto \langle c, \Delta(\theta)\rangle$ over $\mathcal{F}'(\theta)$. We have $C(\theta) \leq C'(\theta)$ from $\mathcal{F}'(\theta) \subset \mathcal{F}(\theta)$ since $\Lambda(\theta) \subset \Lambda'(\theta)$. Now consider $c \in \mathcal{F}(\theta)$ such that $C(\theta) = \langle c, \Delta(\theta)\rangle$, and define $c'$ with $c'(x) = \infty$ if $x = x^\star(\theta)$ and $c'(x) = c(x)$ otherwise. Then we have $c' \in \mathcal{F}(\theta)$ and $\langle c', \Delta(\theta)\rangle = \langle c, \Delta(\theta)\rangle$ so that $C'(\theta) \leq C(\theta)$. Therefore $C'(\theta) = C(\theta)$.

Consider $\theta$ fixed and consider $(\theta^k)_{k \geq 0}$ a sequence in $\Theta$ converging to $\theta$. We will prove that $C(\theta^k) \to_{k \to \infty} C(\theta)$. It is sufficient to prove that $\limsup_{k \to \infty} C(\theta^k) \leq C(\theta) \leq \liminf_{k \to \infty} C(\theta^k)$.

We first prove that $\limsup_{k \to \infty} C(\theta^k) \leq C(\theta)$. By Assumption 1 and 5, there exists $k_0$ such that $x^\star(\theta^k) = x^\star(\theta) \ \forall k \geq k_0$. If $k \geq k_0$ then $\Lambda'(\theta) = \Lambda'(\theta^k)$ by definition of $\Lambda'$. Consider $c^\star$ an optimal solution, that is $c^\star \in \mathcal{F}'(\theta)$ and $C(\theta) = \langle c^\star, \Delta(\theta)\rangle$. Define the sequence $c^k$:

$$c^k = \frac{c^\star}{\inf_{\lambda \in \Lambda'(\theta)} \langle c^\star, D(\theta^k, \lambda)\rangle}$$

We have $c^k \in \mathcal{F}'(\theta^k)$ since $\Lambda'(\theta) = \Lambda'(\theta^k)$ and:

$$\inf_{\lambda \in \Lambda'(\theta^k)} \langle c^k, D(\theta^k, \lambda)\rangle = \frac{\inf_{\lambda \in \Lambda'(\theta^k)} \langle c^\star, D(\theta^k, \lambda)\rangle}{\inf_{\lambda \in \Lambda'(\theta)} \langle c^\star, D(\theta^k, \lambda)\rangle} = 1.$$

We have that $(\theta^k, \lambda) \mapsto D(\theta^k, \lambda)$ is continuous and

$$\inf_{\lambda \in \Lambda'(\theta)} \langle c^\star, D(\theta^k, \lambda)\rangle = \min_{\lambda \in \overline{\Lambda'(\theta)}} \langle c^\star, D(\theta^k, \lambda)\rangle.$$

Also, since $\Lambda'(\theta) \subset \Theta$, and $\Theta$ is compact, we have that $\overline{\Lambda'(\theta)}$ is compact. Therefore, by Berge's maximal theorem:

$$\min_{\lambda \in \overline{\Lambda'(\theta)}} \langle c^\star, D(\theta^k, \lambda)\rangle \to_{k \to \infty} \min_{\lambda \in \overline{\Lambda'(\theta)}} \langle c^\star, D(\theta, \lambda)\rangle \geq 1,$$

since $c^\star \in \mathcal{F}'(\theta)$. Now $c_k \in \mathcal{F}'(\theta^k)$ for all $k$ so that

$$\limsup_{k \to \infty} C(\theta^k) \leq \limsup_{k \to \infty} \langle c^k, \Delta(\theta^k)\rangle \leq \langle c^\star, \Delta(\theta)\rangle$$

We have proven that $\limsup_{k \to \infty} C(\theta^k) \leq C(\theta)$.

We now prove that $\liminf_{k \to \infty} C(\theta^k) \geq C(\theta)$. There exists a sequence $c^k \in \mathcal{F}'(\theta^k)$ such that $C(\theta^k) = \langle c^k, \Delta(\theta^k)\rangle \ \forall k$. We prove that $(c^k)_k$ is bounded by contradiction. If $(c^k)_k$ is

unbounded, then it admits a subsequence $(c^{k_m})_m$ with $||c^{k_m}|| \to_{m \to \infty} \infty$. This readily implies that $\langle c^{k_m}, \Delta(\theta^{k_m})\rangle \to_{m \to \infty} \infty$ since $\Delta(\theta^{k_m}) \to_{m \to \infty} \Delta(\theta)$, and all components of $\Delta(\theta)$ are strictly positive. Therefore $\limsup_{k \to \infty} C(\theta^k) = \infty$, a contradiction since we have proven $\limsup_{k \to \infty} C(\theta^k) \le C(\theta) < \infty$.

Hence $(c^k)_k$ is bounded. Consider $c$ one of its accumulation points and $(c^{k_m})_m$ a subsequence converging to $c$. Since $c^{k_m} \in \mathcal{F}'(\theta^{k_m})$, for any $\lambda \in \Lambda'(\theta) = \Lambda'(\theta^{k_m})$ we have $\langle c^{k_m}, D(\theta^{k_m}, \lambda)\rangle \ge 1$. By continuity of $D$ this implies $\langle c, D(\theta, \lambda)\rangle \ge 1$ for all $\lambda \in \Lambda'(\theta)$ and $c \in \mathcal{F}'(\theta)$. Now $C(\theta^{k_m}) = \langle c^{k_m}, \Delta(\theta^{k_m})\rangle \to_{m \to \infty} \langle c, \Delta(\theta)\rangle \ge C(\theta)$ since $c \in \mathcal{F}'(\theta)$. This holds for all accumulation points so we have proven $\liminf_{k \to \infty} C(\theta^k) \ge C(\theta)$ which concludes the proof of the first statement. The second statement follows directly. $\qquad\square$

## 2.3 Proof of Theorem 2

The proof is articulated around upper bounding the number of times a suboptimal arm $x \ne x^\star$ may be selected in one of the 3 phases of OSSB. We recall the definition of $\Delta(\theta, x) = \mu^\star(\theta) - \mu(x, \theta)$ and further define:
$$\psi(\theta) = |\mathcal{X}| ||c(\theta)||_\infty \sum_{x \in \mathcal{X}} \Delta(\theta, x).$$

Consider $\kappa > 0$, and $\delta(\kappa) > 0$ such that for all $\lambda \in \Theta$ verifying $||\theta - \lambda||_\infty \le \delta(\kappa)$ the following holds:

(i) $C(\lambda) \le C(\theta)(1 + \kappa)$,

(ii) $\psi(\lambda) \le 2\psi(\theta)$,

(iii) $x^\star(\theta) = x^\star(\lambda)$.

From Assumptions 1, 5 and 6 and Lemma 3, the mappings $\theta \mapsto c(\theta)$, $\theta \mapsto C(\theta)$ and $\theta \mapsto x^\star(\theta)$ are continuous, so that such a $\delta(\kappa)$ exists.

**Exploitation Phase.** In this phase, we select arm $x^*(m(t))$, and $N(x,t) \ge c(m(t), t)(1 + \gamma)\ln t$ for all $x$, so:
$$\sum_{x \in \mathcal{X}} N(x,t)D(m(t), \lambda, x) \ge (1 + \gamma)\ln t, \forall \lambda \in \Lambda(m(t)).$$

If a suboptimal arm is selected $x^*(m(t)) \ne x^\star(\theta)$, then we must have $\theta \in \Lambda(m(t))$ so that event $\mathcal{A}(t)$ defined as:
$$\sum_{x \in \mathcal{X}} N(x,t)D(m(t), \theta, x) \ge (1 + \gamma)\ln t.$$

occurs. From Lemma 1 we have
$$\sum_{t \ge 1} \mathbb{P}(\mathcal{A}(t)) \le G(\gamma, |\mathcal{X}|).$$

**Certainty equivalence.** We now upper bound the number of times a suboptimal arm is chosen and $\theta$ is not estimated with sufficient accuracy. Define $\mathcal{B}(t)$ the event that $x(t) \ne x^\star(\theta)$, that we are not in the exploitation phase and that $||\theta - m(t)||_\infty > \delta(\kappa)$. Define $\mathcal{B}(x,t)$ the event that $\mathcal{B}(t)$ occurs and $|\theta(x) - m(x,t)| > \delta(\kappa)$ so that $\mathcal{B}(t) = \cup_{x \in \mathcal{X}} \mathcal{B}(x,t)$. We prove that if $\mathcal{B}(t)$ occurs then $\min_x N(x,t) \ge \frac{\varepsilon s(t)}{2}$. Assume that this is false, then there exists $s(t)/2$ rounds $\{t_1, ..., t_{s(t)/2}\} \subset \{1, ..., t\}$ where $\min_x N(x,t) \le \varepsilon s(t)$. After $|\mathcal{X}|$ such rounds $\min_x N(x,t)$ is incremented by at least 1, so that $\min_x N(x,t) \ge \frac{s(t)}{2|\mathcal{X}|}$. Since $\varepsilon < \frac{1}{|\mathcal{X}|}$ this is a contradiction. Therefore, if $\mathcal{B}(t)$ occurs, then we have both $\min_x N(x,t) \ge \frac{\varepsilon s(t)}{2}$, and $||\theta - m(t)||_\infty > \delta(\kappa)$. By Lemma 2 and a union bound we conclude that:
$$\sum_{t \ge 1} \mathbb{P}(\mathcal{B}(x,t)) \le \sum_{t \ge 1} \sum_{x \in \mathcal{X}} \mathbb{P}(\mathcal{B}(x,t)) \le \frac{2|\mathcal{X}|}{\varepsilon \delta^2(\kappa)}.$$

**Estimation and Exploration Phase.** We are now left with the regret caused by rounds at which $\theta$ is estimated accurately, and we are not in the exploitation phase. Define $\mathcal{C}(t)$ the event that $x(t) \in$

$\underline{X}(t) \cup \overline{X}(t)$ and that $\mathcal{B}(t)$ does not occur. Define the regret caused by such events:

$$W(T) = \sum_{t=1}^{T} \sum_{x \in \mathcal{X}} \Delta(x, \theta) \mathbf{1}\{x(t) = x, \mathcal{C}(t)\}.$$

Assume that $\mathcal{C}(t)$ occurs and that $x(t) = x$. We first upper bound $s(t)$. If $x = \underline{X}(t)$, then $N(x, t) \leq \min_{x \in \mathcal{X}} N(x, t)$. Since we are not in the exploitation phase, there exists $x'$ such that $N(x', t) \leq c(x', m(t))(1 + \gamma) \ln t \leq ||c(m(t))||_\infty (1 + \gamma) \ln T$. Hence $N(x, t) \leq ||c(m(t))||_\infty (1 + \gamma) \ln T$. If $x = \overline{X}(t)$, then $N(x, t) \leq c(x, m(t))(1 + \gamma) \ln t \leq ||c(m(t))||_\infty (1 + \gamma) \ln T$. In both cases $N(x, t) \leq ||c(m(t))||_\infty (1 + \gamma) \ln T$. Since $s(t)$ is incremented whenever $\mathcal{C}(t)$ occurs, we deduce that $s(t) \leq |\mathcal{X}| ||c(m(t))||_\infty (1 + \gamma) \ln T$.

We can now bound the number of times $x$ is selected. If $x = \underline{X}(t)$, we have $N(x, t) \leq \varepsilon s(t) \leq \varepsilon |\mathcal{X}| ||c(m(t))||_\infty (1 + \gamma) \ln T$, and if $x = \overline{X}(t)$ we have $c(x, m(t))(1 + \gamma) \ln T$. We deduce that $\sum_{t=1}^{T} \mathbf{1}\{x(t) = x, \mathcal{C}(t)\}$ is upper bounded by:

$$(c(x, m(t)) + \varepsilon |\mathcal{X}| ||c(m(t))||_\infty)(1 + \gamma) \ln T.$$

Summing over $x$ and using the fact that if $\mathcal{C}(t)$ occurs then $||m(t) - \theta||_\infty \leq \delta(\kappa)$, so that

$$\sum_{x \in \mathcal{X}} \Delta(x, \theta) c(m(t)) \leq C(\theta)(1 + \kappa),$$

$$\varepsilon |\mathcal{X}| ||c(m(t))||_\infty \sum_{x \in \mathcal{X}} \Delta(x, \theta) \leq 2\varepsilon \psi(\theta),$$

we get:

$$W(T) \leq (C(\theta)(1 + \kappa) + 2\varepsilon \psi(\theta))(1 + \gamma) \ln T.$$

Putting everything together we obtain the finite-time regret upper bound:

$$R^\pi(T) \leq \mathbb{E}(W(T)) + \mu^\star(\theta)\Big(\sum_{t \geq 1} \mathbb{P}(\mathcal{A}(t)) + \mathbb{P}(\mathcal{B}(t))\Big)$$

$$\leq \mu^\star(\theta)\left(G(\gamma, |\mathcal{X}|) + \frac{2|\mathcal{X}|}{\varepsilon \delta^2(\kappa)}\right)$$

$$+ (C(\theta)(1 + \kappa) + 2\varepsilon \psi(\theta))(1 + \gamma) \ln T.$$

This implies that:

$$\limsup_{T \to \infty} \frac{R^\pi(T)}{\ln T} \leq (C(\theta)(1 + \kappa) + 2\varepsilon \psi(\theta))(1 + \gamma).$$

The above holds for all $\kappa > 0$, which yields the result:

$$\limsup_{T \to \infty} \frac{R^\pi(T)}{\ln T} \leq (C(\theta) + 2\varepsilon \psi(\theta))(1 + \gamma).$$