[Reviews · NeurIPS 2017]

Reviewer 1



Problem Studied: The paper introduces and studies a wide range of stochastic bandit problem with known structural properties. This includes linear, lipschitz, unimodal, combinatorial, dueling bandits. Results: The paper derives an instance specific regret lower bound based on the structure and gives an algorithm which matches the lower bound. The algorithm needs to solve a semi-infinite LP in the general case, but they show that it can be efficiently done for special cases (which follows from previous literature). Comments: The paper introduces a nice general framework for several of existing known results for several bandit problems such as linear, lipschitz, unimodal, combinatorial, dueling bandits. This encapsulates all these in very broad theoretical framework which can be useful for any new bandit problems.